# Altitudinal variation in reproductive investment among *Gryllus campestris* populations

**David Martínez-Viejo** [1], **Rolando Rodríguez-Muñoz** [2]*, **Alfredo F. Ojanguren** [1]

1 Departamento de Biología de Organismos y Sistemas, Universidad de Oviedo, Oviedo, Spain, 2 Centre for Ecology and Conservation, University of Exeter, Exeter, United Kingdom

* R.Rodriguez-Munoz@exeter.ac.uk

## Abstract

Life history traits determine the organismal abundance within a population and are affected by the presence of trade-offs that modify relationships between traits. These relationships can vary across different environments either by local adaptation or phenotypic plasticity. Reproductive traits have direct fitness implications and therefore are suitable to study among population variation linked to environmental differences. Factors such as altitude are often related to differences in key physical factors like ambient temperature or the subsequent duration of the suitable period for annual activity. The aim of this work was to compare reproductive investment in females of the field cricket *Gryllus campestris* originated from different altitudes, but without identifying the components (genetic vs. phenotypic) of the analysed variables. This species has an annual cycle; after a winter diapause, adults emerge to breed by early mid spring to produce a new generation of nymphs. The study used females collected at the start of the 2021 breeding season, from 10 populations living in the Cantabrian region (Northern Spain). Five of them were located in areas under 170m a.s.l. and the other five above 1100m. Females were allowed to mate with a male from the same population and to lay eggs that we then collected to estimate egg mass and laying rate; both traits were analysed controlling for female size. We found no effect of altitude on any of the three measured traits, female size, egg mass, and laying rate, as well as on the relationships between each pair or traits. Our results suggest that this species is tolerant to environmental variation for the measured traits, showing that it has mechanisms to cope with a range of ambient temperatures.

## Introduction

Life cycles are determined by survival and reproduction, both depending on age and size at birth, maturation, and death [1]. One of the main elements of life history theory is the existence of trade-offs constraining the evolution of fitness related traits [2,3]. Understanding what maintains variation in these trade-offs requires differentiation between phenotypic and genetic variation [4]. Individuals divide available resources into three basic functions: growth,

---

**Data Availability Statement:** Data are accessible from Figshare (DOI 10.6084/m9.figshare. 26982787).

**Funding:** This study was financially supported by UK Research and Innovation (UKRI) Natural

---

Environment Research Council (NERC) [https://www.ukri.org/councils/nerc] in the form of grants (NE/H02364X/1, NE/L003635/1, NE/R000328/1, NE/V000772/1) received by RR-M. No additional external funding was received for this study. The funder had no role in study design, data collection and analysis, decision to publish, or preparation of the manuscript.

**Competing interests:** The authors have declared that no competing interests exist.

somatic maintenance, and reproduction [5]. Two relationships emerge from the patterns of energy expenditure by parents on their offspring. First, when reproductive investment per individual offspring increases, the number of offspring necessarily decreases. Second, if reproductive investment per individual increases, the fitness of each descendant will increase [6].

In oviparous species, egg size can vary according to environmental factors such as quality and quantity of available nutrients, temperature, predation, and male phenotypes [7]. Females that produce higher quality offspring do not inevitably have to sacrifice fecundity as there are alternative mechanisms like increasing egg mass, a higher egg water content to avoid desiccation, or a decline in egg mass with female age [8].

There is evidence of between population variation in the size-fecundity relationship, and it can change according to environmental conditions. Some species adjust eggs size in anticipation of the environment that the juveniles will experience after hatching, mainly the quantity and quality of the food source, predation risk, and temperature [9]. For instance, in years of low food availability, organisms may invest more per offspring as compared to years of higher food abundance [10]. Differences in fitness in relation to individual size seem to exist also due to the higher hatching success of larger eggs, especially in more stressful environments [5]. Theoretical models have concluded that selection favours larger eggs in poorer quality environments, so that offspring increase survival, develop faster, or reach larger sizes at maturity [11].

Consistent patterns of variation in body size and life history traits are found across altitudinal gradients, despite inter-annual variation [12]. Individuals from different environments would show phenotypic differences because of plasticity or adaptation to local conditions [13].

Adaptive phenotypic plasticity arises when the same genotype produces different phenotypes depending on environmental conditions, or due to the evolution of different geographically adapted genotypes [14]. It is important to know whether species with large geographic distributions remain competitive across a range of environments through a generalist phenotype, specialisation by phenotypic plasticity, or local adaptation [15]. Organism size variation is plastic, but also reflects genetic and environmental interactions and covariation [13].

Small ectotherms, such as insects, are highly exposed to climatic variables such as temperature and sun exposure, as they depend on the environment to maintain their temperature [16]. Temperature variation with altitude is an important factor linking growth physiology to behavioural development and population dynamics as it interacts with competition and resource density [12]. This is why altitudinal gradients provide such a powerful research tool, with the potential to provide insights into future impacts of global warming. There are few detailed empirical studies that have attempted to analyse variation in fecundity and egg size across populations from contrasting thermal environments, which is crucial to have accurate models that represent reality [17].

Crickets have been used as models to understand variation in life history traits across thermal gradients [18]. They can be found in very different habitats with highly variable environmental conditions, so populations in these habitats are susceptible to changes in body size, offspring size and fecundity due to low dispersal and isolation of population. While the trade-off between egg size and fecundity is widespread [6], it remains uncertain whether this trade-off has different outcomes across different populations.

In this paper we analyse variation in reproductive investment in field crickets (*Gryllus campestris*) from two contrasting environments. Our aim is to test the hypothesis that there is adaptation to local conditions in relation to the reproductive investment of populations from different altitudes (and therefore ambient temperatures and the annual activity period). We will not be able to separate the genetic and environmental components of phenotypic variance, but if we find a difference, this might be suggestive that there is adaptation or plasticity (either

neutral or maladaptive). We tested whether altitude of origin influences female size, egg mass and egg number, and the relationships between those variables. We expected larger females to lay more eggs rather than larger eggs because eggs mass is expected to be optimised before egg number [6]. However, there may be constraints on egg production leading to variation in both mass and number of eggs. Based on the lower mean temperature of high-altitude locations and their shorter "growing season", we predict that because of the shorter activity season at higher altitude, females from these locations would be smaller and would lay fewer but relatively larger eggs than those from low altitude.

## Material and methods

### Study system

*G. campestris* is a flightless orthopteran living mainly in grassland habitats. In our study area, adults emerge (reach sexual maturity) in early to mid-spring, females mate and keep laying eggs into the ground to early summer. During spring 2021, we trapped adult crickets or last instar nymphs from ten different sites in Asturias and Cantabria (North Spain), using Flipper Traps [19]. Five of the sites were located at altitudes below 170 m and five over 1,100 m above sea level (Fig 1, Table 1). Mean annual temperatures at 70 and 1,300 masl are 14.0 and 10.1˚C, respectively. Mean number of days with maximum temperature below 10˚C are 7.7 and 99 (data extracted from two weather stations located at H07 and near L03, for the period 2021–2023). *G. campestris* is not a protected species in the area, but some of the captures were carried out inside protected areas. To collect cricket in those sites, we got a permit from the Consejería de Medio Rural y Cohesión Territorial of the regional government of Asturias. Different catchment basins act as natural barriers reducing connectivity between sampled populations. During the capture sessions, we stored the crickets in 20 l boxes with egg cardboard for shelter and

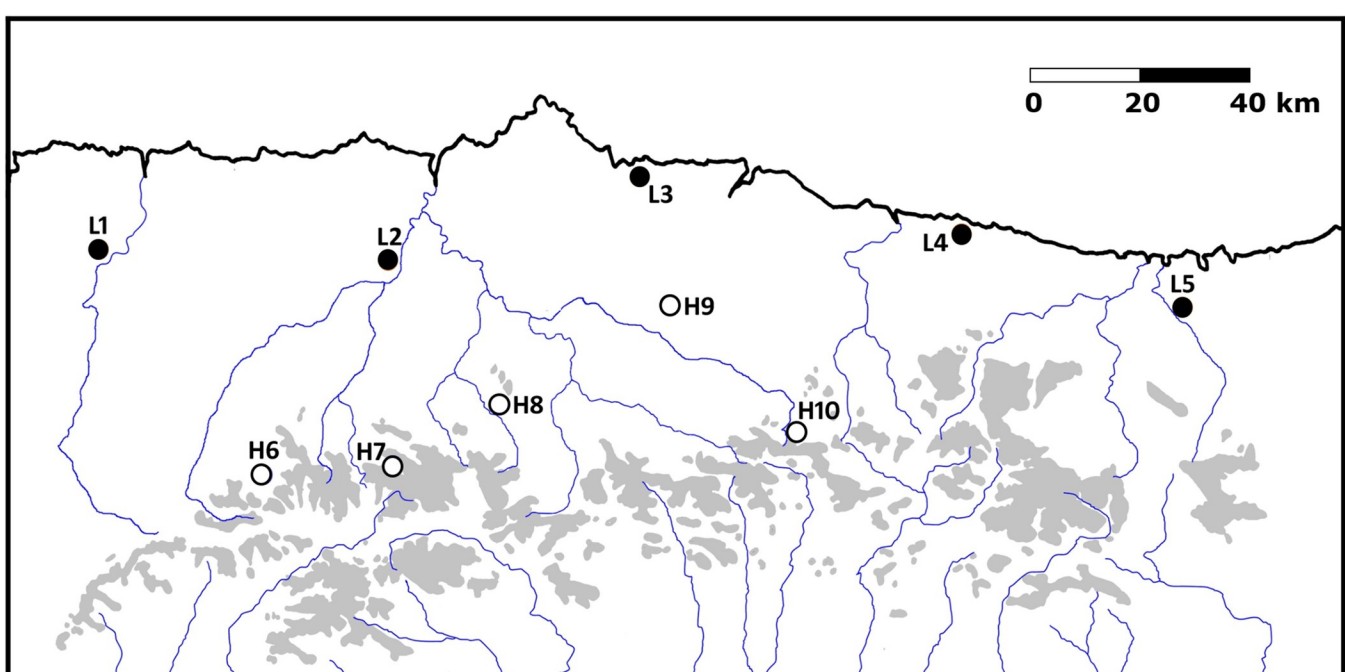

**Fig 1. Location of the collecting sites.** Those coded with L, are located below 170 metres a.s.l.; the ones coded as H are located above 1,100 metres. Grey colour represents areas above 1500 metres and blue lines show the main rivers.

food (standard rodent diet) and water (a 5 ml glass vial filled with water and closed with a cotton plug). We moved them to a laboratory located at 70 masl near population L03 (Fig 1) within 1–2 days of capture.

## Cricket rearing and breeding

At the laboratory, we isolated up to ten males and ten females per population (depending on the numbers available; see Table 1) in 2 l plastic boxes, prioritising adults over nymphs. Each box had a window covered with a thin metal mesh for ventilation, a piece of egg cardboard for shelter, and food and water *ad libitum*. Additional females (for those populations with more than 10 available) were kept in groups of 4–5 using the same type of 2 l boxes, whereas males in excess remained grouped in the 20 l boxes used during capture. We placed the boxes on shelves within building that had sides that were open to the air so crickets were at ambient air temperature and light regime but remained in the shade. The only exception to this rearing protocol was population H07, for which we only captured nymphs. We placed these nymphs in open-topped plastic boxes located outdoors to promote adult emergence under natural conditions. The boxes had an area of 0.2 m$^2$ with a 14 cm layer of soil with natural grass growing on it, and. We used three boxes per sex, with 3–4 females or males in each box, and fed them with sunflower seeds. Once they emerged as adults, they were isolated following the protocol described above for the rest of the populations. Males and females maintained in groups were included in the protocol of isolation and mating (see below), whenever we had to replace any of the originally isolated individuals because of death or failure to mate and/or produce eggs.

Some of the crickets collected as adults, might have mated in the wild before being trapped, and so would be able to lay fertile eggs straightaway. However, it is likely that many of the females we captured were still virgin because they were caught at the start of the breeding season. From mid-May, we carried out mating trials with all the isolated females to make sure that they had mated with at least one male, and so were able to lay fertile eggs. At each mating trial, we put one pair of crickets of the same population within a 0.5 l plastic cup, with a piece of paper on the bottom to provide a gripping surface. We left the pair together in the cup and checked them at intervals of a few minutes over a period of 1–3 hours, to see if a mating had occurred (the spermatophore is visible, attached under her ovipositor). After their first mating trial, we weighed the females, took a photograph for thorax measurement, and moved them to a room indoors with natural light and 25°C temperature, recording which females had mated. In nature, females of this species lay eggs in the ground using their long ovipositor. In the lab,

**Table 1. Location of cricket collection sites in 2021 in North Spain.** Pop shows the ID of each population. Coordinates for each location are included in UTM format. Date shows the date when the crickets were collected at each site. TW, EM and LR, show the number of individuals included in the analysis of thorax width, egg mass and laying rate, respectively.

| Pop | Location | UTM X | UTM Y | Alt | Date | TW | EM | LR |
|-----|----------|-------|-------|-----|------|----|----|----|
| L01 | Trelles | 683731 | 4817456 | 85 | 20-May-2021 | 8 | 10 | 10 |
| L02 | Laneo | 730627 | 4806304 | 66 | 27-Apr-2021 | 4 | 5 | 5 |
| L03 | Trubia | 277565 | 4820128 | 75 | 26-Apr-2021 | 7 | 8 | 8 |
| L04 | Hontoria | 345113 | 4812719 | 46 | 01-May-2021 | 9 | 8 | 9 |
| L05 | Bielva | 381517 | 4795578 | 119 | 05-May-2021 | 9 | 9 | 9 |
| H06 | Brañas de Abajo | 707218 | 4768328 | 1240 | 08-May-2021 | 3 | 4 | 4 |
| H07 | Valle de Lago | 729089 | 4772238 | 1300 | 07-May-2021 | 5 | 5 | 5 |
| H08 | Llanuces | 263209 | 4782845 | 1118 | 05-May-2021 | 7 | 6 | 6 |
| H09 | Peña Mayor | 296857 | 4794393 | 1126 | 06-May-2021 | 4 | 5 | 5 |
| H10 | Tarna | 319128 | 4774872 | 1121 | 09-May-2021 | 4 | 4 | 4 |

we provide them with a small Petri dish (3 cm in diameter and 1 cm in depth) filled with wet river sand. All females that did not mate on any given mating trial were tried again in subsequent days a maximum of 3–5 times. For locations where more females were available, we limited the mating trials to three before discarding a female and trying another. For those with limited number of females, we did up to five mating trials. After those trials, we discarded any female that did not mate in the lab and did not lay any eggs, replacing it with another female from the group boxes if available.

## Data collection

Every 1–3 days, we sieved the sand in each female's dish to collect any eggs and put a new dish with clean sand. We distributed the eggs of each sieved dish on a wet cotton pad, placed in a Petri dish for incubation. We took pictures of each incubation dish to allow subsequent counting of the number of eggs. We also stored a sample of 20 eggs per female in 95% alcohol for subsequent estimation of egg mass. We did not aim to record lifetime fecundity, but to test for differences between different environments. Thus, we estimated laying rate by collecting eggs over a period of a female's life and then divided the total number collected by the duration of that period. This allowed us to analyse potential differences in number of eggs laid between altitudes. For females that started to lay without mating in the lab (indicating that they had mated in the wild), we recorded the start of the egg-laying period on the day of the first mating trial. For those that mated in the lab, we started it from the day of the first successful mating trial. In both cases, we recorded the end of egg-laying on the day when we collected the last sample of eggs.

We used thorax width as an indicator of female size as commonly used for this group [20]. We took photographs of each female from dorsal view and measured the images with the image processing software ImageJ [21]. To calibrate the measurements, we used a reference grid included in each photograph (S1 Fig). To estimate fecundity, we counted the eggs in each batch using the same software used to measure female size.

To measure egg mass we used dry weight in order to remove the potential effect on egg of differences in water content. Eggs were dried for 24 hours in an air heater at 50˚C. In some of the egg samples, 1–2 eggs were broken or with clear signs of deterioration by the time of being weighed. Thus, we actually used three samples of five eggs per female were placed on small aluminium foil portions, previously weighed and with a written code to identify the sample. To ensure that the three samples from each female were independent replicates in relation to the weighing procedure, we weighed them on 3 different days. Samples were weighed using a laboratory scale with an accuracy of 10-μg.

## Statistical methods

We carried out statistical analyses in R (v4.0.3; [22]) under Rstudio (v. 1.3.1093) using the packages *lme4* (v1.1.28; [23]) and *lmerTest* (v3.1.3; [24]). We included population as a random factor; our replicate populations provide independent samples within altitudes, but possible differences among them were not of specific interest in this study.

To explore differences in female size (thorax width), egg mass (dry weight) and laying rate (mean number of eggs laid per day) between altitudes, we used mixed models with population as a random factor nested within altitude. We used the Satterthwaite method to cope with the imbalance in sample sizes among populations and between altitudes. We extracted type III ANOVA results using the *anova* function.

For egg mass and laying rate, models included standardized female size as a covariate, and its interaction with altitude. Although the duration of the laying period varied among females,

there was no difference between altitudes that could introduce a bias ($F_{1,\ 6.3}$ = 0.684, $P$ = 0.438). We ran the following models:

Thorax width ~ Altitude + (1 | Population)
Egg mass ~ Thorax width * Altitude + (1 | Population)
Laying rate ~ Thorax width * Altitude + (1 | Population)

## Results

Adult emergence is quite synchronised in *G. campestris* populations, and most adults emerge within 2–3 weeks [25]. Although we do not know the emergence date of each experimental cricket, the fact that we found nymphs in all the high altitude populations and three out of five in the low altitude, indicates that the oldest individual in the experiment was no more than 2–3 weeks old. Part of the females that refused to mate in the lab laid viable eggs (16% and 25% of the high and low altitude samples, respectively), showing that they had mated in the field. We included 68 females in the analyses, with the laying period lasting 20.1 ± 8.4 days. Our data reveal a high inter- and intra-population variability (Fig 2); overall, thorax width

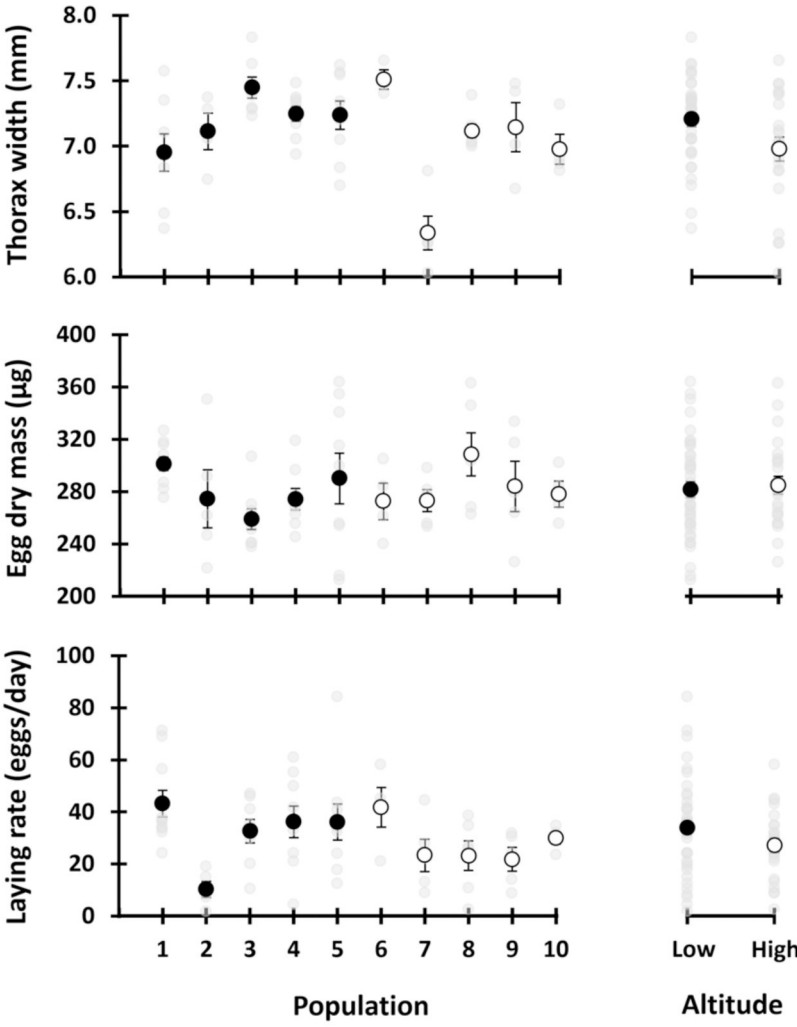

**Fig 2. Mean and standard error of each trait.** Populations on the left, and clustering populations according to altitude on the right (Low, black points, populations below 170 m; High, white points, populations above 1100 m). Data points shown as light grey points in the back.

**Table 2. Output of the *anova* function for three mixed models analysing the effect of altitude on female size, egg mass and laying rate (size included as a covariate for the latter two variables).** Significant P values highlighted in bold italics.

| Dependent | Fixed effects | Sum of squares | NumDF | DenDF | F | P |
|---|---|---|---|---|---|---|
| Female size | Altitude | 0.069 | 1 | 7.6 | 0.901 | 0.372 |
| Egg mass | Thorax width | 5903.8 | 1 | 31.8 | 5.699 | ***0.023*** |
| | Altitude | 77.0 | 1 | 5.6 | 0.074 | 0.795 |
| | Thorax width*Altitude | 4310.2 | 1 | 31.8 | 0.299 | 0.588 |
| Laying rate | Thorax width | 289.6 | 1 | 31.9 | 1.311 | 0.261 |
| | Altitude | 145.8 | 1 | 6.1 | 0.660 | 0.447 |
| | Thorax width*Altitude | 12.7 | 1 | 31.9 | 0.057 | 0.812 |

(mean ± standard deviation) was 7.12 ± 0.38 mm, egg dry weight 282.8 ± 36.0 μg and laying rate 31.5 ± 16.9 eggs/day. We found no difference between altitudes in any of the three variables, thorax width, egg mass and laying rate (Table 2). Thorax width had a weak positive effect on egg size, although this relationship did not differ between altitudes (Tables 2 and 3; Fig 3). We did not find any relationship between female size and laying rate (Table 2).

## Discussion

Theory predicts that individuals from different environments will show phenotypic differences as a consequence of genetic or adaptive changes [13]. Our aim was to test whether living in different thermal environments, influences adult size and the patterns of reproductive investment in a model ectotherm. Mean annual ambient temperature is lower in high than in low altitudes, so the portion of the year with temperatures above the development threshold, is reduced. In ectotherms, this means a generally lower growing temperature and shorter duration of the period when animals can growth and reproduce [26].

The way these altitude effects can influence reproductive traits is determined by a number of common trade-offs and their relationship with fitness. Bigger females lay bigger eggs [5], and bigger eggs lead to bigger juveniles, which have higher chances to survive [27]. However, there is a compromise between the size and the number of eggs that a female can produce [6]. Under this scenario, we would expect adult females to be smaller at high altitudes, and to lay larger eggs to increase offspring survival [5,27]. As a consequence of that increase in egg size, we would expect a decrease in fecundity.

None of these predictions were met in our study, as we found no differences in female or egg size and fecundity due to altitude. Not finding a difference does not mean that it does not exist, but the fact that we were far from detecting any effect, suggests that if there is a difference it is probably small. It could be argued that the period the crickets spent in a common

**Table 3. Output of a mixed model analysing the effect of altitude on egg mass in the field cricket *G. campestris*, with female size as a covariate.** Females collected from ten populations (random effects) located at five high and five low altitude sites. Significant P values highlighted in bold italics.

| Fixed effects | Estimate | SD | df | t | P |
|---|---|---|---|---|---|
| Intercept | 281.3.43 | 11.34 | 6.5 | 24.795 | ***< 0.001*** |
| Thorax width | 16.84 | 9.66 | 21.6 | 1.743 | 0.096 |
| Altitude Low | -4.22 | 15.47 | 5.6 | -0.273 | 0.795 |
| Thorax width*Altitude Low | -6.28 | 11.48 | 31.8 | -0.547 | 0.588 |
| **Random effects** | **Variance** | **SD** | | | |
| Population | 398.1 | 19.95 | | | |
| Residual | 1036.0 | 32.19 | | | |

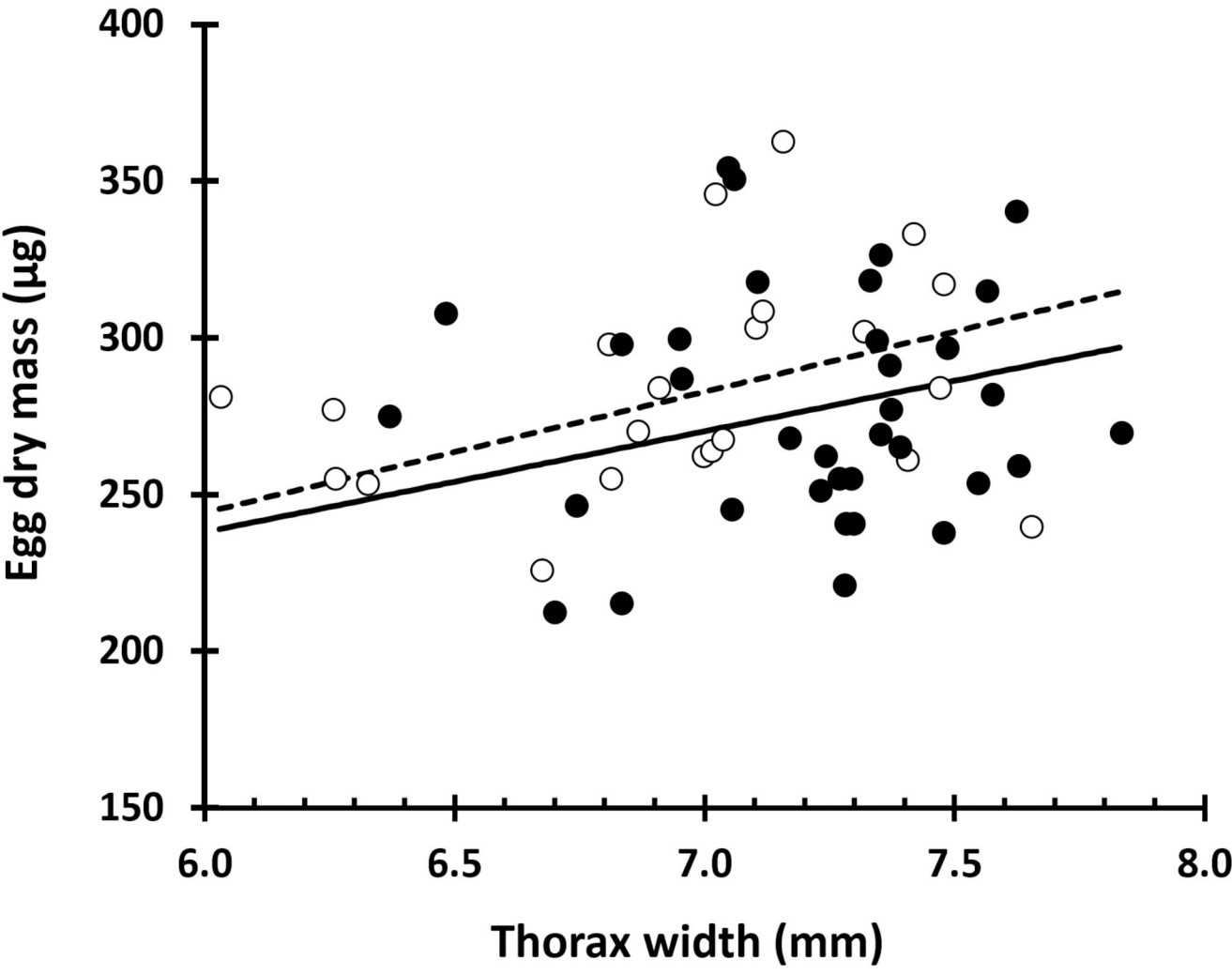

**Fig 3. Relationship between female thorax width and egg dry mass in the field cricket *Gryllus campestris*.** Data source from 10 different populations (not identified in the plot), five of them living above 1,100 masl (broken line and black dots) and five living below 170 masl (continuous line and white dots). Regression lines have been produced using the *R* package *effects* from a mixed model ran using the *lme4* package, including population as random effects, and thorax width and altitude (and their interaction) as fixed effects.

environment in the lab, in a low altitude location, could have influenced the reproductive output of the high altitude females. The most relevant influence could happen if there was a countergradient variation effect [28] that could underlie the similarity between altitudes. We think that if there is any effect due to that common lab period it would be very small, as by that time of the year environmental differences are not very big. During that lab period, and in order to promote mating and egg laying, we kept the experimental crickets at a temperature that we think is near their optimum at that stage of their life cycle, and that does not seem to differ among crickets living at different altitudes [29]. *G. campestris* uses behavioural thermoregulation to control its body temperature by sun basking, which allows these crickets to increase it up to 20°C above the air temperature [30]. This means that during the reproductive season, crickets have the possibility to overcome the air temperature differences between altitudes by modulating the time they spend basking. Another potential effect that could influence egg laying in our study would be the difference in phenology between the high and low altitudes. If

crickets from low altitude would have had more time to mate in the wild before being caught, and number of matings would increase egg laying, this could influence the number of eggs collected in the lab in relation to altitude. Although we know adult emergence happens slightly later in the year at high altitude, we found nymphs in three of the five low altitude sites during the collection dates, indicating that the reproductive season was still starting also in them. Even if there were some more females already mated in the low than in the high altitude, that difference would not be expected to be very high. Also, unless there is countergradient variation, the effect of that difference would be opposite to what we found, i.e. there should have been a higher fecundity in low altitude as compared to high.

Our results suggest that the factors influencing fecundity and female and egg size might not differ between the two studied environments, which does not mean that different adaptations could exist at other stages of the life cycle where both altitudes have more contrasted conditions. Evidence of thermal adaptation in this species in the same area has been reported in previous studies [15], which showed that nymphs from high altitude are able to recover faster than those from low altitude, under extreme cold events. Solar radiation, higher in mountain areas, is a key factor for our study species that is known for using basking as a mechanism for behavioural thermoregulation, in common with other insects [30,31]. Crickets can use basking as a way to overcome the lower ambient temperature during reproductive season, when sun is more frequent. Moreover, the existence of variation in female size among populations, but not between altitudes, might be indicative that there are other environmental factors that influence how big crickets can grow at different sites.

We found that bigger females lay bigger rather than more eggs (see Fig 3). The existence of a compromise between the size and number of propagules [6] has been supported across many different taxa [32], including other studies on *Gryllus* [17]. In our study, females laying bigger eggs did not show lower fecundity. Other studies have found that females producing high quality offspring do not necessarily have to sacrifice fecundity. There are alternative mechanisms that allow females laying larger eggs to reduce their lifetime fecundity. These include increasing reproductive effort, a higher egg water content, or a decrease in egg size with female age [8].

Climate change has started to have strong effects across ecosystems and species. These changes are clearly visible but there is a shortage of studies quantifying life history traits and physiology of species to predict whether and how they will adapt to these changes [33]. In ectotherms, increasing temperature is likely to accelerate life cycles by producing shorter juvenile stages, increasing reproductive investment, and reducing maintenance investment [34]. In the context of global change and rising temperatures, there will be taxa and species that will increase and expand, while others will be severely affected [35]. Depending on time per generation, population size or the amount of genetic variation within the population, adaptive evolution may occur fast enough as to make individuals resilient to the environmental changes [35]. In crickets, an increase in temperature within the physiological range of a species has been shown to produce an increase in egg size and number, as well as faster embryo development [36]. However, global change will affect many other variables, not just temperature. These might include changes in water availability, vegetation cover parasite and predator communities and numerous other variables [37]. Most taxa undergo range shifts to compensate for increased temperatures due to climate change, increasing their optimal living elevation [38] or latitude [39]. Climate change affect differently at low and high altitudes, and this will cause problems for individuals migrating along altitudinal gradients, this could also apply to latitudinal migrants [40].

In our study species, sun is an important factor to control body temperature, so in cold areas, changes in cloud cover over the year could be even more important than changes in air temperature. Because insects are a critical part of ecosystems, a better understanding of the

potential effects of climate change on their populations is essential and require further study [41]. Under this changing environment, plastic organisms that adapt faster to environmental changes will have a greater chance of survival [35]. Studies on plasticity and evolution across populations living in contrasting environments, aimed to predict responses to climate change, would require the study plasticity and evolution across populations.

## Supporting information

**S1 Fig. Photograph of a cricket female placed in a groove to avoid the animal to move, showing the identity label and the reference grid used for calibration.** The yellow line represents the distance measured as thorax width.
(TIF)

## Acknowledgments

We thank Nete Randlev and Kristian Riis for helping with cricket catching in the wild. Tom Tregenza and Paul Hopwood made useful comments on the manuscript. Luis Fernando Alonso Sierra helped to get the required permit from the Consejería de Medio Rural y Cohesión Territorial of the regional government of Asturias in due time.

## Author Contributions

**Conceptualization:** Rolando Rodríguez-Muñoz, Alfredo F. Ojanguren.

**Data curation:** David Martínez-Viejo.

**Formal analysis:** Rolando Rodríguez-Muñoz.

**Funding acquisition:** Rolando Rodríguez-Muñoz.

**Investigation:** David Martínez-Viejo.

**Methodology:** Rolando Rodríguez-Muñoz.

**Project administration:** Rolando Rodríguez-Muñoz.

**Resources:** Rolando Rodríguez-Muñoz.

**Supervision:** Rolando Rodríguez-Muñoz, Alfredo F. Ojanguren.

**Writing – original draft:** David Martínez-Viejo.

**Writing – review & editing:** Rolando Rodríguez-Muñoz, Alfredo F. Ojanguren.

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
