## [Decision Letter · Decision Letter 0]

8 Mar 2024

PONE-D-24-00871Altitudinal variation in reproductive investment among Gryllus campestris populationsPLOS ONE

Dear Dr. Rodríguez-Muñoz,

Thank you for submitting your manuscript to PLOS ONE. After careful consideration, we feel that it has merit but does not fully meet PLOS ONE’s publication criteria as it currently stands. Therefore, we invite you to submit a revised version of the manuscript that addresses the points raised during the review process.

**Both reviewers raised some concerns and provide several suggestions to improve the manuscript. One of the main issues relates to the experimental design (and the fact that all individuals were reared under the same environmental conditions) and the interpretation of the results. Please take into account the reviewer’s comments and carefully revise the manuscript before resubmitting.**

We look forward to receiving your revised manuscript.

Kind regards,

Nicoletta Righini, PhD

Academic Editor

PLOS ONE

Journal Requirements:

NERC An individual-level approach to understanding responses to climate in wild ectotherms  (Climate & fitness): NE/V000772/1

Reviewers' comments:

Reviewer's Responses to Questions

**Comments to the Author**

1. Is the manuscript technically sound, and do the data support the conclusions?

Reviewer #1: Partly

Reviewer #2: No

2. Has the statistical analysis been performed appropriately and rigorously? 

Reviewer #1: Yes

Reviewer #2: Yes

3. Have the authors made all data underlying the findings in their manuscript fully available?

Reviewer #1: No

Reviewer #2: No

4. Is the manuscript presented in an intelligible fashion and written in standard English?

Reviewer #1: Yes

Reviewer #2: Yes

5. Review Comments to the Author

**Reviewer #1: **The submitted manuscript describes an experiment examining local adaptation in field cricket life history traits across an elevational gradient. The study collected adult and late-stage juveniles from five low-elevation and five high-elevation populations that were each separated by geographic barriers. Mature crickets were kept under common laboratory conditions. After mating, the study measured female body size, egg size, and egg laying rate. The study tested for elevational differences in female body size, along with the relationship between female body size and egg traits to test if reproductive investment strategies have evolved across elevations. The results are interpreted in the context of insect responses to climate change. Overall, the study is straightforward, and most of the methods are clearly articulated. The statistical analyses are elegant and justified. I have a few comments about limitations in the experimental design and interpretation of the results that I believe should be addressed in the manuscript before publication. However, I am generally supportive of the manuscript, and I believe a revision could meet the criteria for publication in the journal.

Comments:

1) The manuscript finds that phenotypes do not differ between low and high elevation populations. However, because crickets from each population were not reared under common or reciprocally transplanted conditions, the study cannot elucidate whether this lack of a phenotypic difference is due to (A) a lack of evolutionary change, (B) a lack of phenotypic plasticity, or (C) countergradient selection. The manuscript mentions in the introduction (line 71) that phenotypes could differ among environments due to evolved and/or plastic responses. However, the conclusions and abstract should clearly state that this study does not reveal the source of phenotypic similarities between elevations.

2) Similarly, there is well-established and taxonomically pervasive evidence that insect life histories often evolve and/or show phenotypic plasticity in response to environmental factors correlated with elevation (temperature, hydration, oxygen), making the study’s findings surprising. I believe many researchers would hypothesize that countergradient selection underlies the phenotypic similarities observed among these cricket populations. The Introduction and Discussion would be greatly improved by discussing this hypothesis. Conover & Schultz 1995 TREE (doi: 10.1016/S0169-5347(00)89081-3) would be a helpful reference for this discussion. A main goal of the manuscript is to interpret the results in the context of climate change responses. The patterns of evolution and/or plasticity that underly the phenotypic similarities observed here are important for determining how egg size strategies will respond to climate change in different populations, and this should be addressed in the discussion. For instance, if phenotypic similarities are due to a lack of evolutionary change and a lack of developmental plasticity across elevations, then the phenotypes will likely be robust in the face of climate change. However, if phenotypic similarities stem from countergradient selection, then climate change could still alter cricket life history traits.

3) The study focuses on elevational differences in temperature when interpreting the results. Populations at different elevations experience differences in a number of environmental factors, including hydration and oxygen availability. The introduction and discussion would benefit from a more balanced discussion of these differences beyond temperature. Furthermore, more details regarding how laboratory conditions compared to environmental conditions in the low and high elevation populations should be included in the main text. Is 25C more representative of low or high elevation populations, or is it in-between?

4) The manuscript should provide more information about how many females from each population were known to have mated in the field (for example, in Table 1). It appears that low elevation populations mature earlier, and therefore are more likely to have females that mated in the field. If females mated multiple times in the field, then that could generate elevation differences (or similarities) in egg fertilization/laying. The manuscript would therefore be greatly improved by including supplemental analyses that remove data from females known to have mated in the field.

5) The effect estimates in Table 3 seem very high, indicating that the data may not be standardized relative to the mean and standard deviation for these analyses. If not already done, then standardizing the data in this way is more appropriate for evaluating effect size (see Schielzeth 2010, Methods Ecology & Evolution, doi: 10.1111/j.2041-210x.2010.00012.x). This procedure should not change the reported P values.

6) It is not clear why the sample sizes for populations L02 and L03 are not available if these populations were included in the analyses.

7) Following the journal’s guidelines for data availability, the authors should deposit relevant data in a public data repository or provide the data in the manuscript, rather than have data available upon request. Figure 2 could also be improved by adding the data points behind the means and standard errors.

**Reviewer #2:** Rev PONE D-24-871

l38 …both traits were analysed controlling for female size.

180-81 … such as temperature…

l88 delete “faithfully”

l100-101 it is not clear until you read the methods why “laying rate”, I suggest to simply say “egg number” at this point

l101 We expected…

l106 …would be smaller and would lay…

l107-109 Delete last sentence in intro, this can be mentioned in the discussion.

l116 the abstract said under 170 masl and here 200 m, chose one for consistency.

l126 why are number of captured crickets “not available” for two sites? Report the numbers that were used for analyses. Even if more were captured, they are irrelevant; if not explain why?. Why are there two sites labeled as L05? If they represent a single site then merge them in only one line in the table. “XYZ” column tittles for coordinates and altitude seem too cryptic, label with more details and include units (UTM coordinates, m asl). Could you provide some columns summarizing relevant climate variables for each site such as mean temperature, average date of first and last below 0ºC temperature in the year, average rainfall, degree days or anything you think could relate to reproductive performance. It would be good to see if altitude differences really represent a sizable climate variation, and these data are easy to get nowadays for almost any point in the world, certainly for Northern Spain

1122 Review map, after converting coordinates for L1 and H10 and projecting on Google Earth, the coastline does not match and the scale that should represent 20 km is actually 33 km. See attached png file.

l132 space between 2 and l?

l137-138 Where was this building geographically compared to the collecting sites? Comparing the climate regime of the lab to the collecting sites may be useful for interpretation of results as this is where females ended reproductive maturity and spent most of their egg laying period, thus reproductive “choices” may have been greatly influenced by these conditions, and since they were all in the same conditions, this may a feasible explanation for the results of the study.

162 “A maximum” can only have one value, 3 or 5? If the max value differs between samples, explain why.

l169 why about 20 eggs and not simply exactly 20 eggs each time. Were these eggs taken randomly throughout the period? Could egg size differ at different periods of the reproductive period thus?

l202 if female size were perfectly correlated with egg mass, thorax width would increase as a linear function but egg mass as a cubic function, so it is better to use cubic thorax width as a covariate and in all analyses involving egg mass at least.

l217 was laying rate normally distributed? Rates frequently are not. Only if it was mean and a standard deviation values would be meaningful.

Otherwise use medians and quartiles.

l219 use “egg mass” instead of “egg size” since the former is the actual operational value you measured

l230 remove the word “Low” from the table as you are presenting results for the “Altitude” variable

l239-334 I provide general comments for the discussion. No specific comments on the text are provided as I did for the previous sections since I suggest that the whole discussion should be rewritten due to two main reasons:

The design and interpretation of results assumes that any difference between high and low populations would be maintained outside their natural climate environment, in other words that there would be no plasticity in reproductive output even though phenotypic plasticity is mentioned in the introduction, thus it should be considered in the discussion of the results as well. Since all individuals were reared under the same environmental conditions after being collected and throughout the last part of their developmental period in many cases, and most of their reproductive period, they had plenty of time to “adjust” to the experimental rearing conditions, possibly explaining the lack of significant effects of altitude. It would be ideal to also measure reproductive output if maintained in their original condition, and compared to those that were placed under the new experimental conditions, or have actually crossed conditions (high altitude moved to low altitude conditions and vice versa plus the appropriate control groups), but the present experimental design opens the possibility of plastic adjustment to experimental conditions but offers no way to assess if indeed it happened.

Non-significant results can not be treated with the same logic of significant results, in other words, lack of significant differences can not be directly translated into no biological difference since there is always the possibility of low statistical power, among other reasons. There is extensive literature discussing this topic and the authors should refer to it, just to give 2 examples I offer these references, buth there are many others:

https://www.ncbi.nlm.nih.gov/pmc/articles/PMC4114196/

https://online.ucpress.edu/collabra/article/3/1/9/112371/Too-Good-to-be-False-Nonsignificant-Results

Most of the discussion is built on the assumption that these results are solid evidence of no biological difference between high and low populations, when they are only statistically non significant differences, which is VERY different.

6. PLOS authors have the option to publish the peer review history of their article (what does this mean?). If published, this will include your full peer review and any attached files.

Reviewer #1: No

Reviewer #2: **Yes: **Rogelio Macías-Ordóñez

---

## [Author Response · Author response to Decision Letter 0]

26 Jun 2024

This has been uploaded in the attach files section

---

## [Decision Letter · Decision Letter 1]

2 Aug 2024

PONE-D-24-00871R1Altitudinal variation in reproductive investment among Gryllus campestris populationsPLOS ONE

Dear Dr. Rodríguez-Muñoz,

Thank you for submitting your manuscript to PLOS ONE. After careful consideration, we feel that it has merit but does not fully meet PLOS ONE’s publication criteria as it currently stands. Therefore, we invite you to submit a revised version of the manuscript that addresses the points raised during the review process.

I agree with Reviewer 1 in that the ms has greatly improved and the authors responded adequately to the reviewers' queries. Reviewer 1 points out a couple of minor issues; once these are addressed, the ms can be accepted for publication.

We look forward to receiving your revised manuscript.

Kind regards,

Nicoletta Righini, PhD

Academic Editor

PLOS ONE

Journal Requirements:

Reviewers' comments:

Reviewer's Responses to Questions

**Comments to the Author**

1. If the authors have adequately addressed your comments raised in a previous round of review and you feel that this manuscript is now acceptable for publication, you may indicate that here to bypass the “Comments to the Author” section, enter your conflict of interest statement in the “Confidential to Editor” section, and submit your "Accept" recommendation.

Reviewer #1: (No Response)

2. Is the manuscript technically sound, and do the data support the conclusions?

Reviewer #1: Partly

3. Has the statistical analysis been performed appropriately and rigorously? 

Reviewer #1: Yes

4. Have the authors made all data underlying the findings in their manuscript fully available?

Reviewer #1: Yes

5. Is the manuscript presented in an intelligible fashion and written in standard English?

Reviewer #1: Yes

6. Review Comments to the Author

Reviewer #1: The revision does great job addressing my comments on the original submission. The methods and tables have been clarified, and the data points added to Figure 2 go a long way to improve the interpretation of the results. My concerns regarding conceptual limitations in the interpretation have been addressed in extensive changes to the introduction and discussion. I only have two comments that I think still need be addressed:

1) The introduction shouldn’t state that the study attempts to test predictions of local adaptation. The prediction in line 100 is currently speculative (“if we find a difference, this might be suggestive that there is adaptation”), but ultimately, adaptation can’t be inferred by looking only at phenotypic differences/similarities. Any phenotypic differences could instead represent neutral/maladaptive plasticity. At the same time, countergradient variation is local adaptation that results in phenotypic similarities between populations. The rest of the predictions laid out in the last few sentences of the introduction are appropriate.

2) The manuscript now appropriately states that it can’t parse genetic/environmental sources of phenotypic similarities/differences. However, this also means that the manuscript should be more speculative about what the results tell us about insect responses to climate change. I previously mentioned that if phenotypic similarities are due to a lack of evolution or plasticity, then lines 306 and 322 are correct in suggesting that some insect life history traits may be robust in the face of climate change. However, if the phenotypic similarities are due to countergradient variation, life history traits in these populations should instead show plasticity in response to climate change. I think these two statements should be removed (lines 306 and 322). The manuscript could just explain that future studies examining plasticity and evolution across populations are needed to predict responses to climate change.

7. PLOS authors have the option to publish the peer review history of their article (what does this mean?). If published, this will include your full peer review and any attached files.

Reviewer #1: No

---

## [Author Response · Author response to Decision Letter 1]

24 Sep 2024

We have made all the requested changes

---

## [Editor Report · Decision Letter 2]

30 Sep 2024

Altitudinal variation in reproductive investment among Gryllus campestris populations

PONE-D-24-00871R2

Dear Dr. Rodríguez-Muñoz,

We’re pleased to inform you that your manuscript has been judged scientifically suitable for publication and will be formally accepted for publication once it meets all outstanding technical requirements.

Kind regards,

Nicoletta Righini, PhD

Academic Editor

PLOS ONE

---

## [Editor Report · Acceptance letter]

15 Oct 2024

PONE-D-24-00871R2 

PLOS ONE

Dear Dr. Rodríguez-Muñoz, 

I'm pleased to inform you that your manuscript has been deemed suitable for publication in PLOS ONE. Congratulations! Your manuscript is now being handed over to our production team.

Kind regards, 

on behalf of

Dr. Nicoletta Righini 

Academic Editor

PLOS ONE